# Effect of Special Low-Protein Foods Consumption in the Dietary Pattern and Biochemical Profile of Patients with Inborn Errors of Protein Metabolism: Application of a Database of Special Low-Protein Foods

**DOI:** 10.3390/nu15153475

**Published:** 2023-08-06

**Authors:** Dolores Garcia-Arenas, Blanca Barrau-Martinez, Arnau Gonzalez-Rodriguez, Rafael Llorach, Jaume Campistol-Plana, Angeles García-Cazorla, Aida Ormazabal, Mireia Urpi-Sarda

**Affiliations:** 1Nutrition, Food Science and Gastronomy Department, Xarxa d’Innovació Alimentària (XIA), Faculty of Pharmacy and Food Science, Food Science and Nutrition Torribera Campus, University of Barcelona, Av. Prat de la Riba 171, 08921 Santa Coloma de Gramenet, Barcelona, Spain; dolores.garciaa@sjd.es (D.G.-A.);; 2Inborn Errors of Metabolism Unit, Sant Joan de Déu Hospital, Passeig Sant Joan de Déu 2, 08950 Esplugues de Llobregat, Barcelona, Spain; 3Institute for Research on Nutrition and Food Safety (INSA-UB), Universitat de Barcelona, Av. Prat de la Riba 171, 08921 Santa Coloma de Gramenet, Barcelona, Spain; 4Centro de Investigación Biomédica en Red Fragilidad y Envejecimiento Saludable (CIBERFES), Instituto de Salud Carlos III, 28029 Madrid, Spain; 5Metabolic Unit, Neuropaediatrics Department, Sant Joan de Déu Hospital, Passeig Sant Joan de Déu 2, 08950 Esplugues de Llobregat, Barcelona, Spain; 6Centro de Investigación Biomédica en Red de Enfermedades Raras (CIBERER), Instituto de Salud Carlos III, 28029 Madrid, Spain; 7Clinical Biochemistry Department, Sant Joan de Déu Hospital, Passeig Sant Joan de Déu 2, 08950 Esplugues de Llobregat, Barcelona, Spain; 8Institut de Recerca Sant Joan de Déu, Santa Rosa 39-57, 08950 Esplugues de Llobregat, Barcelona, Spain

**Keywords:** inborn errors of intermediate protein metabolism, special low-protein foods, low-protein diet, database

## Abstract

In inborn errors of intermediate protein metabolism (IEM), the effect of special low-protein foods (SLPFs) on dietary intake has been scarcely studied. The aim of this study was to compare the nutritional profile of SLPFs with usual foods and to assess whether their intake determines the dietary pattern and affects the plasma biochemical profile in children with IEMs with different protein restrictions. A database with the nutritional composition of 250 SLPFs was created. A total of 59 children with IEMs were included in this cross-sectional observational study. The greatest significant differences in macronutrient composition were observed between dairy, meat, fish, and egg SLPFs and regular foods. After stratifying subjects by SLPFs, the participants with the highest intake (>32%) had a higher total energy intake and lower intake of natural protein than those in the lowest tertile (<24%) (*p* < 0.05). However, when stratifying subjects by dairy SLPF intake, children in the highest tertile (>5%) showed a higher intake of sugars, total and saturated fats, and higher plasma levels of total and low-density lipoprotein cholesterol than those in the first tertile (<1%) (*p* < 0.05). The variability in the nutritional composition of SLPFs highlights the need for up-to-date databases which would greatly assist in optimizing individualized recommendations for children with IEMs and protein restrictions.

## 1. Introduction

Disorders of protein and amino acid metabolism, such as phenylketonuria (PKU), maple syrup urine disease (MSUD), tyrosinaemia (TYR), homocystinuria (HCU), organic acidaemias (OAAs), and urea cycle disorders (UCD), are a group of inherited metabolic conditions caused by the deficiency of enzymes or transporters involved in amino acid metabolism. Clinical phenotypes range from asymptomatic to life-threatening [1]. Due to the deficiency, toxic metabolites accumulate and affect various organs such as the brain, liver, and kidney. These metabolic decompensations can lead to serious neurocognitive problems. Therefore, a basic principle of management for these disorders is to reduce the concentrations of toxic substrates in tissues and plasma by minimizing the intake of nutrients that produce them [2].

In most of these pathologies, the primary treatment approach is dietary modification [3], in addition to pharmacological interventions. Dietary treatment involves restricting natural protein intake contained mainly in foods such as meat, fish, eggs, dairy products, legumes, nuts, and also in cereals, pasta, and bread [4]. The amount of natural protein is determined individually based on the severity of the condition, age, rate of growth, and metabolic status. Thus, it is essential to avoid and minimize the consumption of protein-containing foods in the diet of these children due to the detrimental effects associated with their intake. The consumption of such foods can lead to the accumulation of harmful substrates, thereby exacerbating the condition. For example, in the case of PKU, elevated levels of phenylalanine can occur, while patients with UCD may experience an accumulation of ammonium. These metabolic imbalances can lead to progressive neurological deterioration and manifest various symptoms, such as lethargy, seizures, headaches, vomiting, and growth failure [2,5,6].

Regular monitoring of plasma amino acid levels is necessary to ensure that they remain within optimal ranges or to adjust vitamin or cofactor supplementation to increase enzyme activity when residual enzyme activity remains [4]. In addition to protein restriction, treatment often involves supplementation with precursor-free L-amino acid supplements (PFAAs) [7] or essential amino acid supplements [5], consumption of special low-protein foods (SLPFs), and the incorporation of naturally low-protein foods such as fruits, vegetables, and fats. These dietary interventions complement the daily diet and are essential for achieving and maintaining good metabolic control and preventing neurological deterioration due to metabolic decompensation [6].

The dietary approach for managing these metabolic disorders typically involves calculating and measuring the natural protein source, adding carbohydrates and fats to the diet to increase caloric intake and prevent catabolism, and supplementing with calculated amounts of PFAAs as needed, along with SLPFs that provide energy and variety to the diet [8]. SLPFs are considered essential in managing these disorders, as they not only meet energy needs but also help maintain anabolism and improve the variety of the diet, thereby helping to maintain metabolic control within target ranges [9]. SLPFs are processed foods that have limited proteins, with their composition primarily based on carbohydrates and fats. Their flavour and aesthetic properties are prioritized over their nutritional composition, which may differ from analogous products containing natural protein [10,11,12]. A study by Pena et al. [10] found that the availability of SLPFs varies across different European countries including Portugal (*n* = 73), Belgium (*n* = 92), Italy (*n* = 256), and Germany (*n* = 94), while 146 products were reviewed in the United Kingdom [11]. However, their availability was unknown in Spain, Denmark, and the Netherlands, among others [10]. Although nutritional information for some of these products is available in metabolic nutritional programs such as ODIMET [13], MetabolicPro [14,15], or the Metabolic Diet App [15], no databases are currently available in Spain, and the nutritional quality of SLPFs has not been compared to that of regular foods.

The nutrients provided by each group of foods in a healthy diet are known; however, the contribution of different food groups and the consumption of SLPFs in the total diet of individuals with inborn errors of intermediary protein metabolism (IEM) with restricted protein diets are not clear. In the case of phenylketonuria, it has been estimated that the intake of SLPFs ranges between 30% and 60% of total calories [10,11,12]. However, for other protein-restricted amino acid conditions, this has not been well studied, and it is likely to depend on the severity of the protein restriction and tolerance to natural protein.

It has been reviewed by Verduci et al. [16] that PKU children who were compliant with the diet did not seem to exhibit different cardiovascular risk factors compared to healthy subjects. However, several studies in adult PKU patients have suggested a potential risk for cardiovascular diseases due to several factors, such as lower HDL-C or higher homocysteine [17], higher BMI in certain groups of PKU patients [18], and elevated levels of inflammatory and oxidative stress markers [19], among others. The quality and compliance of the dietary treatment could be some of the factors related to the future evolution of the IEM patient. However, to our knowledge, there are no studies evaluating SLPF intake in the plasma glucose and lipid profile. Thus, our hypothesis is that a high consumption of SLPFs may potentially impact the diet quality in individuals with IEMs and could lead to alterations in their lipid and glucose profiles. Such changes in lipid and glucose profiles might act as triggering factors for future cardiovascular diseases in this specific population.

Therefore, the aim of this study is to compare the nutritional profile of SLPFs with that of regular foods and to evaluate the extent to which the intake of SLPFs determines dietary patterns and plasma lipid and glucose profiles in a group of children with IEMs with different degrees of protein restriction.

## 2. Materials and Methods

### 2.1. Special Low-Protein Foods (SLPFs) vs. Regular Foods

A total of 250 SLPFs were collected from Spanish manufacturers and suppliers, and an in-house database was created. Products were divided into 15 categories and included milk replacers; ice cream/whipped cream; cheese; breakfast cereals; bread and mixes; flour; pasta; rice; cookies; cakes and desserts; chocolate; ready meals; and meat, fish, and egg replacers.

The nutritional information was collected from the mandatory nutritional labelling of the products or from technical data sheets when available. Nutritional data were obtained per 100 g or 100 mL of products for: energy; protein; total carbohydrate; sugars; fibre; total fat; saturated fatty acids (SFAs); and sodium or salt. Following the criteria of Wood et al. [11], when nutritional data were lower than a certain value, i.e., ‘<0.1′ or ‘<0.5′, a subtraction of 0.001 was applied to these numbers and values of ‘0.099′ or ‘0.499′ were used. The mean and standard deviation values for every nutrient were calculated. The nutritional profile was compared with a total of 247 homologous products with proteins (regular foods) using the mean of values of the nutritional information from three open-access food composition databases: the Spanish Food Composition Database: BEDCA [20,21], the CESNID food composition tables [22], Denmark’s Frida Food Data [23], and from two nutritional programs: ODIMET [13] and DIAL^®^ [24], containing more than 800 foods. Only 38% of the SLPFs included information on some vitamins and minerals, so they could not be compared with their respective regular protein-containing products.

### 2.2. Subjects and Study Design

This was a cross-sectional observational study conducted from January 2021 to March 2022. Eligible patients for this study were children diagnosed with IEMs by newborn screening or genetically confirmed, treated with protein-restricted diets, aged 10 months to 17 years, without a language barrier or difficulty of understanding, and managed at Congenital Metabolic Disease Unit of Sant Joan de Déu Hospital (HSJD), as well as who continuously attended clinical appointments. In total, 59 children (27 girls and 32 boys) were recruited. Of these children, 30 had phenylketonuria (PKU); 9 had PKU tetrahydrobiopterin responders (PKU-BH4); 2 had maple syrup urine disease (MSUD); 4 had classical homocystinuria (HCU); 4 had organic acidaemias (OAAs) (which included 1 methylmalonic acidaemia (MMA); 1 glutaric aciduria type 1 (GA1) and 2 with 3-hydroxy-3-methylglutaric aciduria (3-HMG)); 4 had urea cycle disorders (UCDs) (which included 2 argininosuccinate synthetase deficiency (ASS), 1 with ornithine transcarbamoylase deficiency (OTC) and 1 with hyperornithinaemia, hyperammonaemia, and homocitrullinuria (HHH)); 3 had hereditary tyrosinaemia type I (HTI) and 3 had a liver transplant including 2 with deficiency of argininosuccinate lyase (ASL) and 1 with propionic acidaemia (PA).

The study was approved by the Research Ethics Committee (CEIm) of the Fundació de Recerca Sant Joan de Déu (codeCEIm PIC-63-21). The parents or legal caregivers for all children provided their written informed consent before their inclusion in the study.

### 2.3. Measurements

Weights and lengths for children under 2 years of age were obtained by standard techniques using digital baby-weighing scales and crown–heel length on a scaled length board. Heights and weights of children older than 2 years were measured using a combined stadiometer and a digital weight-measuring station (Seca 284, Seca, Hamburg, Germany). Height to the nearest 0.1 cm and weight to the nearest 0.1 kg were recorded. Anthropometric measurements were expressed as age- and sex-specific Z-scores, using the SEGHNP Paediatric Nutritional Application, available online: https://www.seghnp.org/nutricional (accessed on 31 May 2022), based on the Barcelona Longitudinal Growth Study 1995–2017 [25].

Parents completed a 3-day food record (3DFR) of their children, including one nonworking day (i.e., Saturday or Sunday). The trained metabolic dietitian calculated calories and macro- and micronutrient intake of the 3DFR using the nutrient analysis software program ODIMET [13], which is an extensive nutrient database of amino acid formulas and supplements and SLPFs, and also allows the inclusion of the SLPFs recruited in our study. Plasma biochemical parameters were also collected. Blood samples were collected by venipuncture after fasting for at least 12 h. The following plasma parameters were determined: total cholesterol; high-density lipoprotein cholesterol (HDL-C); low-density lipoprotein cholesterol (LDL-C); triglycerides (TAGs); and fasting glucose.

### 2.4. Statistical Analyses

Statistical analyses were performed using SPSS version 27.0 software (SPSS Inc., Chicago, IL, USA). The Mann–Whitney non-parametric unpaired test was used to compare energy, nutrients (proteins, total fat, SFAs, carbohydrates, sugars, fibre), and sodium contents from SLPFs and regular foods. These comparisons were presented as a mean and standard deviation (SD).

The anthropometric parameters, dietary intake, and plasma biochemical data were skewed (Kolmogorov–Smirnov and Levene tests) and the natural logarithm of the variables did not normalize them; therefore, the median (interquartile range, IQR) was used to describe them. Spearman’s correlation was used to study the relationship between variables. Participants were also categorized based on tertiles of SLPF consumption and tertiles of dairy SLPF consumption. To study the differences across tertiles, data were analysed using the Kruskal–Wallis and Mann–Whitney post hoc tests or a Chi-square test. In addition, post hoc power analysis was used to calculate the statistical power of total cholesterol and LDL-C parameters according to tertiles of dairy SLPF consumption [26,27]. When data were categorized by disorders of protein and amino acid metabolism, the median (min and max) was used to describe them. A *p*-value < 0.05 was considered significant.

## 3. Results

### 3.1. Comparison of Nutritional Composition between SLPFs and Regular Foods

Table 1 shows the values of the means and standard deviation of energy, protein, total fat, SFAs, carbohydrates, sugars, fibre, and sodium of the SLPFs and a comparison with their respective regular food groups. Thirty-three per cent of the SLPFs were significantly different in energy compared to regular products. The egg replaced group presented significantly higher energy (2.5-fold change) than regular foods. The cheese SLPFs conferred significantly lower energy (0.7-fold change) than regular cheese. The other foods with significant differences showed lower values than 1.06-fold change. As expected, all SLPFs contained significantly lower protein content than regular foods. A total of 40% of food groups showed significant differences in total fat and 33% in SFAs. Cheese SLPFs had significantly lower total fat (0.8-fold change) but higher SFAs (1.1-fold change) than regular cheese. The ‘bread and mixes’ SLPFs group showed higher total fat (1.5-fold change) and SFAs (2-fold change) than regular ‘bread and mixes’, and pasta SLPFs presented significantly lower total fat (2-fold change) and higher SFAs (1.5-fold change) than regular pasta. In addition, breakfast cereal SLPFs and flour SLPFs presented significantly lower fat than regular cereals in terms of a threefold change and a twofold change, respectively. Finally, a higher change in total fats was observed in eggs. Egg SLPFs showed a significant 22-fold change in lower total fat than regular eggs. Significant differences were shown in carbohydrates and sugars in 67% and 40% of food groups, respectively. The groups of cereals (flour, pasta, cookies, cakes), protein replacer groups (meat, fish, and egg), and dairy (milk and cheese) SLPFs contained significantly higher amounts of carbohydrates than regular food. The major differences were observed in the groups of egg replacers (101-fold change but as a trend to significance), meat replacers (20-fold change), cheese replacers (18-fold change), and fish replacers (11-fold change). Some of these SLPFs, jointly with the low-protein flour, reported a higher significant sugar content than regular foods: fish replacers (tenfold change), meat replacers (threefold change) and flour (sixfold change).

From the nutritional analysis, the fibre values were obtained for all the groups except for dairy SLPFs due to unavailability of this parameter in technical data sheets. Some 60% of food groups showed significant differences in fibre. The most significant groups were meat, egg and fish replacers, which contained more fibre than regular foods (260, >100, 60-fold changes, respectively) since regular products had no fibre content. In terms of sodium evaluation, 53% of the groups showed significant differences between regular foods and SLPFs. While significant higher levels of sodium were observed in meat replacers, low-protein pasta, flour and cheese, and a trend in fish replacers, there were other food SLPF groups—such as milk replacers, bread, cookies, and cakes—with significant lower amounts than regular foods.

Therefore, globally, it has been observed that the SLPF groups of cheese, meat, fish, and egg replacers had higher differences in macronutrients with those of regular foods than the other food groups.

### 3.2. Study Population

A total of 59 infants with a median age of 8.9 years were recruited. Table 2 illustrates the characteristics of participants with amino acid metabolism disorders. The main disorders were PKU (51%) followed by PKU-BH4 (15%). All individuals had normal values for weight Z-score and height Z-score, but the BMI Z-score of the participants was normal in 66%. In this regard, 19% were overweight, 12% obese, and 3% underweight, according to Carrascosa and Mesa [25]. After the stratification of patients according to individual acidaemia diseases (Appendix A), the nine participants who showed a BMI Z-score of between 1.0 and 1.49 were those with PKU (*n* = 7), MSUD (*n* = 1), and a PA liver transplanted (*n* = 1); the five subjects who showed a BMI Z-score of between 1.5 and 1.95 were subjects with PKU (*n* = 1) and HTI (*n* = 1); the seven subjects who showed a BMI Z-score greater than 2 were subjects with PKU (*n* = 5), PKU-BH4 (*n* = 1), and MSUD (*n* = 1). On the other hand, one participant with an ASL liver transplanted showed a BMI Z-score of between −1 and −1.49; three participants showed a BMI Z-score of between −1.5 and −1.95, two participants with PKU and one with OAA; and two participants in total, one with PKU-BH4 and one with HCU, had a BMI Z-score lower than −2.0, indicating underweight.

The median and IQR of plasma glucose, triglycerides (TAGs), total cholesterol (TC), high-density lipoprotein cholesterol (HDL-C), and low-density lipoprotein cholesterol (LDL-C) were within the normal range in the complete study population (Table 2) considering reference values [28,29,30]. However, when stratified by pathologies (Appendix A), the results showed that one participant with PKU-BH4 had higher plasma glucose levels (128 mg/dL), indicating diabetic levels (>125 mg/dL), and three subjects had values between 100 and 125 mg/dL, indicating prediabetic levels [30] and they included one patient with HCU, one with OAA, and one with PKU. After the evaluation of TAG values, 27% of participants had values higher than those that were acceptable (Table 2). It was observed that five participants under 9 years old had borderline values of TAGs (75–99 mg/dL) which included one individual with PKU-BH4, one with HCU, one with MSUD, one with HTI, and one with OAA. Furthermore, two subjects with PKU-BH4, one with HTI, and one each with PA and with ASL—both with their liver transplanted—had high values of TAGs (≥100 mg/dL). A total of four participants between 10 and 19 years old had borderline values of TAGs (90–129 mg/dL) and this included three with PKU and one with OAA; in addition, two subjects with PKU had high values of TAGs (≥130 mg/dL) [28,29]. On the other hand, one participant with PKU-BH4, six with PKU, and one each with OAA and ASL—both with their liver transplanted—had borderline total cholesterol values (Table 2 and Appendix A); additionally, one participant with UCD, one with a PA liver transplanted and two with OAA had higher cholesterol values (≥200 mg/dL). In this regard, while 78% and 85% of participants had acceptable ranges of HDL-C and LDL-C, respectively, 10% of subjects presented HDL-C values lower than 40 mg/dL (four with PKU, one with PKU-BH4 and one with HTI) and 3% and 12% of participants conferred LDL-C values higher than 130 mg/dL [22,23] (one with OAA and one with PA and their liver transplanted) and LDL-C values in borderline values (one participant with UCD, two with OAA, two with PKU, one ASL with their liver transplanted and one with PKU-BH4), respectively.

### 3.3. Assessment of Energy and Macronutrient Intake Considering Consumption of SLPFs and Precursor-Free L-Amino Acid Supplement

Table 3 illustrates the characteristics of dietary intake of all participants with disorders of amino acid metabolism. Results showed that the participants followed healthy dietary recommendations [31] considering the percentage of total carbohydrates, total protein, and total fats. However, the percentage of natural protein had the highest variability because it depended on the tolerance of each subject. On the other hand, the percentage of sugars was a bit higher than the 10% recommended [32], and far from the suggested reduction to less than 5% of total energy intake for additional health benefits [33]. A total of 68% of our participants complied with the recommended intake of SFAs (<10%) [32], but only 27% of them achieved the recommended intake of PUFAs (5–10%) [31,32,33,34].

The results showed that 29% of the energy of the dietary intake occurred from the intake of SLPFs, while the 26% comes from the PFAAs intake, which both resulted in 55% of the energy of the total diet. Overall, and as expected, SLPF consumption made a very low contribution to protein intake compared to PFAAs, which provided the highest protein content for patients. It has to be emphasized that similar amounts of carbohydrates were provided by the SLPFs (21%) and from regular foods (22%) (dietary intake minus SLPFs minus PFAAs) (Table 3). Otherwise, 73.5% of sugars came from the diet, since only 17.6% and 8.8% of total sugars were derived from SLPFs and PFAAs, respectively. Participants followed the fibre recommendations [31]. According to the assessed intakes, both natural foods, mainly fruits and vegetables, provided the total dietary fibre of our population. SLPFs contributed fibre to a lesser extent since this information could be more limited on the technical data sheets of the products. SLPFs supplied 7.6% of total fat, with SFAs being the main contributor. However, the intake of PFAAs provided more MUFAs and PUFAs to the participants than SFAs (Table 3).

Appendix A showed total dietary intake, SLPF consumption, and PFAA intake, respectively, stratified by the specific disorder of protein and amino acid metabolism (AA diseases). This was to highlight that patients with MSUD, OAA, and those who had received a liver transplant had the highest percentage of SLPF energy, with values of 31%, 34%, and 33%, respectively. The rest of the IEM groups show that subjects with UCD, and those with PKU and HCU, consumed 18% and 30% SLPFs, respectively. In terms of PFAA intake, the highest percentage was observed in participants with HTI, followed by the PKU and MSUD groups with 42%, 29%, and 27%, respectively. This higher intake was due to their low tolerance to natural protein, which required them to meet their protein needs through PFAA consumption. In contrast, the intake of PFAAs in the HCU, PKU-BH4, and OAA groups fluctuated between 19%, 16%, and 13%, respectively. Moreover, patients with UCD and liver transplantation who did not use PFAAs had a variable intake of SLPFs, with 18% and 33%, respectively, which was also dependent on their tolerance to natural protein.

### 3.4. Relationship among SLPF Consumption, Dietary Intake, Anthropometric, and Plasma Biochemical Parameters

In this study, we mainly focused on dietary intake, and anthropometric and plasma biochemical parameters across tertiles of percentage of SLPF consumption (Appendix A). Participants across tertiles had similar age and Z-scores of weight, height, and BMI. Across tertiles of SLPFs, participants had no significant differences in percentages of total protein consumption, carbohydrates, or fats. However, subjects in the two highest tertiles of SLPF consumption had a higher total energy intake and a lower natural protein intake than participants in the lowest tertile. Participants in the two lowest tertiles of SLPF consumption had significantly higher PUFA consumption compared to the third tertile. It was noted that the percentage of PFAA consumption was higher in the two lowest tertiles of SLPFs, although only significant differences were observed between the second and third tertile. In addition, the levels of plasma glucose, triglycerides, cholesterol, HDL-C, and LDL-C did not change across tertiles of SLPF consumption.

However, significantly positive correlations were observed between participants who consumed more energy from SLPFs, and the ingested carbohydrates, sugars, and fats from them (Table 4). As expected, the consumption of SLPFs was inversely correlated with natural protein, which could indicate less utilization of these SLPF products and more protein tolerance. Also, it was observed that the intake of energy from SLPFs was inversely correlated with total carbohydrates and fats from PFAAs (*p* < 0.05) and with total energy from PFAAs (*p* = 0.05). In addition, the consumption of SLPFs was positively correlated with the intake of protein from PFAAs. This could indicate that subjects that used more SLPFs could need the PFAAs in order to fulfil their protein requirements.

The relationship between carbohydrates, sugars, total fats, and SFAs derived from SLPFs and the other dietary components was also evaluated (Table 4). It is to be highlighted that carbohydrate consumption from SLPFs was inversely correlated with natural protein and total fats in the diet and positively correlated with total dietary carbohydrates. Carbohydrates from SLPFs were positively correlated with sugars, total fats, SFAs, MUFAs, and PUFAs (*p* < 0.05), and total fats from SLPFs were positively correlated with carbohydrates, sugars, SFAs, MUFAs and PUFAs (*p* < 0.05).

However, in the first part of this work, it was concluded that SLPF dairy, meat, fish, and egg replacers had higher differences in macronutrients than the other food groups. In this regard, the consumption of dairy SLPFs made a high contribution to the dietary patterns of subjects, showing a mean (SD) consumption of 5 ± 6%, while the consumption of SLPF meat, fish, and egg replacers made a very low contribution to their diet, with mean (SD) values of 1 ± 2% of total diet, which was a very low percentage of the total diet. Therefore, the results could be difficult to interpret. In this context, the dietary intake, anthropometric, and plasma biochemical parameters were only analysed considering the consumption of dairy SLPFs. In Table 5, it can be observed that participants in the third tertile of dairy SLPF consumption showed a higher consumption of total SLPFs and a lower consumption of PFAAs compared to participants in the first tertile. Participants in the highest tertile also had a higher intake of sugars, total fats, and SFAs than those in the first tertile. In addition, positive correlations were observed between the percentage of energy from the consumption of dairy SLPFs and dietary sugars (r = 0.27, *p* = 0.04), total fats (r = 0.32, *p* = 0.01), and SFAs (r = 0.56, *p* < 0.001).

Significant results were also observed in the plasma biochemical parameters. Subjects in the third tertile of dairy SLPFs had higher plasma total cholesterol and LDL-C than subjects in the two lowest tertiles, and these relationships were supported by positive correlations (r = 0.31, *p* = 0.02 and r = 0.31, *p* = 0.02, respectively). In addition, the ingested sugars through dairy SLPFs were also correlated with the plasma total cholesterol and LDL-C of subjects (r = 0.35, *p* = 0.01 and r = 0.31, *p* = 0.02, respectively). Furthermore, the intake of total fats and carbohydrates from dairy SLPFs was correlated with plasma total cholesterol (r = 0.28, *p* = 0.03 and r = 0.27, *p* = 0.04, respectively).

It is interesting to note that when these results were stratified by plasma reference values (Appendix A), participants with acceptable levels of plasma total cholesterol and LDL-C showed a significant higher plasma total cholesterol and LDL-C in the third tertile of dairy SLPF consumption compared to participants in the first and second tertile (*p* < 0.05). It is worth highlighting that the two individuals who had high plasma levels of LDL-C also had higher consumption of dairy SLPFs.

## 4. Discussion

As far as we know, for the first time, this study compares the nutritional composition of Spanish SLPFs with that of normal protein foods. It is worth noting that, currently, there are no available databases in Spain of these specific products. In this regard, few studies have reported nutritional differences between SLPFs and normal protein foods [10,11,12]. In our study, similar to the findings of Pena et al. [10] and Wood et al. [11], the nutritional profile of SLPFs was significantly different from that of regular foods. Specifically, we found that 67% of the SLPFs had higher carbohydrate content than regular foods, even basic foods in the diet such as low-protein pasta, milk, and cheese replacers. These results are consistent with those of other studies [10], which also reported that the main energy difference between SLPFs and non-SLPFs was due to the higher carbohydrate content [10,11]. Our findings on sugar content are also in line with those of Wood et al. [11], who described a higher percentage of sugars in the meat and fish groups. With regard to total fat, the study by Pena et al. [10] mentioned higher contributions in 58% of SLPFs, especially in the bread group. These results differed from Wood et al. [11], who only reported 37% more fat in some groups such as bread, prepared foods, or milk replacers, among others; in addition to 35% of these, they contained more SFAs compared to foods with normal protein content. Our results revealed few significant variations in the total fat and SFA content between SLPFs and regular foods. Only cheese, bread, pasta, and rice SLPFs had a higher content of SFAs than regular foods, although this difference was limited to a maximum of a 2.5-fold change. Pena et al. [10] also observed higher levels of total fat and SFAs in bread SLPFs. However, egg replacers presented lower SFA levels. This is in line with Wood et al. [11], who reported that 50% of the SLPF subgroups, including flour, cakes, eggs, and fish substitutes, contained less SFAs.

In our study, we identified significant nutritional differences between SLPFs and regular foods, particularly in the categories of dairy products, meat, fish, and egg replacers. However, the findings of the previous study conducted by MacDonald et al. [35], which examined the average amount of SLPFs stored in households for consumption among individuals with PKU, depicted that the most commonly consumed items were low-protein pasta, pizza crusts, sausage/burger mix packets, and cookies. On the other hand, the study by Daly et al. [12] revealed that the most-consumed staple low-protein foods among children aged 5 to 16 with PKU were bread and pasta. This was followed by milk replacers, although in children ≥12 years old the intake was much lower than in children aged <11 years. Additionally, Wood et al. [36] conducted research on NHS spending and the consumption of SLPFs in England, finding that the most frequently prescribed SLPFs were bread, pasta/rice, flour, and milk replacers. These results are consistent with our study, as we observed a higher contribution of dairy replacers to the diet of our patients (5 ± 6%) in comparison to SLPF meat, fish, and egg replacers (1 ± 2%). The variability in the nutritional composition of SLPFs highlights the need for up-to-date databases, which would greatly assist in optimizing individualized recommendations for children with IEM and protein restriction.

To our knowledge, this is the first study investigating the eating behaviour of children with IEMs and the impact of SLPFs on dietary patterns and biochemical profile, taking into account protein restrictions and how they could affect acquired and modifiable risk factors that exacerbate these hereditary conditions.

SLPFs are an essential component of dietary treatment, contributing a significant portion of energy intake (median 29%, IQR range 20–36%) in our patients. This is consistent with the findings of Daly et al. [12], who reported a 30% intake of SLPFs in patients with PKU. However, in other works, a higher percentage (50%) has been reported in patients with PKU [37]. To our current knowledge, the contribution of SLPFs to total diet has not been studied in other protein-related IEMs.

The results of our study indicated that the percentage of total sugars consumed by the population exceeded the recommendations [32], although they consumed the recommended amounts of total carbohydrates. Nevertheless, we found a significant correlation between the percentage of carbohydrates in the total diet and the consumption of carbohydrates from SLPFs (r = 0.35, *p* < 0.05), as well as a positive correlation between sugars in the total diet and sugars from dairy SLPFs (r = 0.28, *p* < 0.05). High sugar consumption has been linked to an increased likelihood of overweight and obesity among youth, regardless of other dietary or macronutrient intakes [38] and alterations of the biochemical profile, such as glucose and triglyceride levels [39]. In a study by Moretti et al. comparing the dietary patterns of individuals with PKU to healthy controls, higher carbohydrate intake, glycemic index, and glycemic load were noted in the PKU population [37]. Interestingly, their PKU population showed lower total and LDL cholesterol levels but higher triglycerides than healthy children. In line with that, we observed a positive trend between total dietary sugar intake and triglyceride levels (r = 0.25, *p* = 0.05; data not shown). However, despite calculating the contribution of SLPFs in the dietary pattern, Moretti et al. did not investigate the association of their consumption with plasma glucose metabolism or lipid profile parameters [37]. On the other hand, Couce et al. found normal glucose levels for nearly all patients, similar to our study. However, they identified altered fasting insulin levels in 26% of the PKU patients. Notably, they saw a significant positive correlation between carbohydrate intake and basal insulin, HOMA-IR, and waist circumference [40]. Interestingly, the study by Evans et al. [41] showed that in patients with PKU from the age of one year, the energy provided by carbohydrates is higher than that of healthy controls [41]. However, although we did not perform a comparison with a control group, all pathology groups in our study had a similar consumption of carbohydrates, with the MSUD group having a slightly higher consumption.

On the other hand, our results suggest that fibre intake is adequate. This is mainly due to the fact that a significant proportion of our patients (68%) consume between 400 g and 600 g of fruits and vegetables per day, as recommended [32]. The consumption of fruits and vegetables has been associated with numerous health benefits such as protecting colonic gastrointestinal health and reducing the risk of cardiovascular disease, among others [42,43]. Interestingly, fibre intake was found to be negatively correlated with total cholesterol levels (r = −0.29, *p* = 0.03), highlighting another potential benefit of consuming these foods in reducing cardiovascular risk [43]. Despite the adequate fibre intake and its beneficial effects, Verduci et al. noted that in individuals with PKU, the low-protein diet led to an increase in carbohydrate intake, as well as higher glycemic index and glycemic load of their diet. This diet could result in a different quality of substrates for microbial fermentation, leading to reduced butyrate production and lower microbial diversity [44]. Bassanini et al. also identified that the quality and quantity of carbohydrate ingested had impacts on the microbiome of PKU individuals, as they observed a depletion of the microorganism *F. prausnitzii*, which is considered a biomarker of health status and a butyrate producer [45]. In our study, the starch content of SLPFs could not be determined due to the lack of nutritional information in technical data sheets but it is used as an ingredient in SLPFs. It is known that the starch in SLPFs is derived from refined sources, such as isolated manioc or potato starch, which have different physiologic properties compared to the complex starch forms found in regular foods [46]. Foods containing refined starches may have a higher glycaemic index, which, when combined with a sedentary lifestyle, could lead to an increased risk of cardiovascular diseases [47]. It is important to highlight the study conducted by De Oliveira et al., which demonstrated that the microbiome of PKU patients exhibited fewer genes involved in starch and sucrose metabolism, as well as glycolysis/gluconeogenesis, compared to healthy subjects. These differences may impact the production of short-chain fatty acids in PKU patients, consequently affecting gut health [48]. In light of these findings, it would have been highly insightful to explore whether individuals who consume higher quantities of SLPFs or dairy SLPFs exhibit distinct microbiota compared to those with lower consumption. Further investigations are necessary to delve deeper into this aspect.

Furthermore, we observed that individuals who consumed more than 5% of their energy from dairy SLPFs also had higher dietary SFAs and sugars. These could be possible factors that potentially contribute to elevated levels of total cholesterol (TC), as well as LDL-C, in the blood. Thus, excessive consumption of SFAs, added sugars (such as sucrose or high fructose corn syrup), and refined carbohydrates leads to changes in LDL-C, HDL-C, and triglyceride, which can increase, and has been associated with increased morbidity and mortality from cardiovascular diseases, both in adults and children [49,50,51,52]. According to the European Food Safety Authority (EFSA), saturated and trans-fat intake are the main dietary determinants of high plasma LDL-C concentrations [53] and the relationship between SFA consumption and plasma cholesterol increase in children is well known [54]. Therefore, monitoring the consumption of SLPFs is crucial, as some of these foods contain significant amounts of SFAs and carbohydrates, like starch or free sugars, compared to regular foods and are widely consumed. Having access to accurate and comprehensive nutritional information on SLPFs is crucial for healthcare professionals because they could provide effective dietary advice to patients and optimize dietary recommendations for patients, considering the potential impact of SLPFs on their overall health and wellbeing.

However, our study has some limitations. One these is the incomplete nutritional information available for some SLPFs, including information about the kind of starch or fibre they contain. The other limitation is that our sample was not homogeneous for the different AA diseases, so it is not possible to compare the dietary pattern between them. The sample size is also limited; however, the statistical power for the main results is adequate since the estimated effect size for plasma total cholesterol levels assessed through dairy SLPF consumption indicated a statistical power of 86%. In addition, the estimated effect size for LDL-C assessed through dairy SLPF consumption revealed a statistical power of 83%. Moreover, the difficulty of working with the paediatric population and the low prevalence of these diseases make our findings particularly relevant and representative. Therefore, our study serves as an important example of the quality of the diet required for managing IEMs and highlights the need for further research in this area.

## 5. Conclusions

Our results suggests that the nutritional composition of SLPFs is different from that of usual foods with protein and their contribution between 18% and 30% to the total diet of our participants has not been related to protein restriction, nor to the excessive consumption of SFAs or carbohydrates. However, a consumption of more than 5% of some SLPFs, such as dairy products, could imply the intake of more sugars, total fats, and SFAs, which could affect the cholesterol levels of these children. Our study highlights the importance of regularly evaluating the protein-controlled diet in IEM in clinical practice to ensure adequate intake of SLPFs and prevent imbalances that could impact the lipid and glucose profile of these patients. As a result, they could prevent potential future complications, including cardiovascular risk factors.

## Figures and Tables

**Table 1 nutrients-15-03475-t001:** Comparison of the nutritional composition between SLPFs and regular foods ^a^.

Food Group	Regular Food or SLPFs	Food Groups	Energy (kcal)	Proteins (g)	Total Fat (g)	SFAs (g)	CHO g)	Sugar (g)	Fibre (g)	Na (mg)
Dairy products	Regular food	Whole/semi-skimmed milk (*n* = 11)	55 ± 9	3 ± 0.2	3 ± 1	2 ± 0.6	5 ± 0.2	5 ± 0.3	0.0 ± 0.0	46 ± 5
	SLPFs	Milk replacers (*n* = 10)	58 ± 16	0.2 ± 0.1 **	3 ± 1	2 ± 0.5	7 ± 2 **	4 ± 2	0.3 ± 0.4 * (*n* = 4)	30 ± 23 *
	Regular food	Ice/whipped cream (*n* = 7)	235 ± 67	3 ± 0.8	16 ± 13	10 ± 8	19 ± 11	17 ± 10	0.3 ± 0.4	50 ± 26
	SLPFs	Ice/whipped cream (*n* = 4)	309 ± 136	1 ± 1 *	14 ± 2	11 ± 5	45 ± 38	28 ± 24	3.2 ± 0.0 (*n* = 2)	167 ± 131
	Regular food	Cheese (*n* = 28)	368 ± 57	25 ± 8	29 ± 5	18 ± 3	1 ± 1	0.7 ± 1	--	737 ± 348
	SLPFs	Cheese (*n* = 38)	272 ± 25 **	0.5 ± 0.6 **	22 ± 3 **	20 ± 3 *	18 ± 6 **	0.4 ± 1	--	828 ± 230 *
Cereals	Regular food	Breakfast cereals ^U^ (*n* = 14)	364 ± 41	8 ± 3	3 ± 2	0.6 ± 0.5	75 ± 14	24 ± 12	6 ± 7	436 ± 357
	SLPFs	Breakfast cereals ^U^ (*n* = 9)	381 ± 5	0.4 ± 0.3 **	1 ± 0.4 *	0.8 ± 0.5	92 ± 3 **	28 ± 13	2 ± 2 *	63 ± 34 †
Bread and mixes	Regular Food	Bread and mixes (*n* = 19)	296 ± 68	9 ± 2	4 ± 5	1 ± 2	55 ± 8	3 ± 2	3 ± 2	583 ± 193
	SLPFs	Bread and mixes (*n* = 33)	295 ± 75	0.7 ± 0.3 **	6 ± 5 *	2 ± 3 *	57 ± 15	4 ± 2 *	6 ± 4 *	381 ± 232 *
Flour	Regular Food	Flour (*n* = 18)	345 ± 13	10 ± 2	2 ± 0.8	0.3 ± 0.1	72 ± 4	0.8 ± 0.7	6 ± 4	63 ± 185
	SLPFs	Flour (*n* = 14)	340 ± 35	0.4 ± 0.2 **	1 ± 1 *	0.5 ± 0.5	81 ± 12 *	5 ± 3 **	3 ± 3 * (*n* = 12)	107 ± 154 **
Pasta	Regular food	Pasta (*n* = 32)	351 ± 8	12 ± 0.8	2 ± 0.4	0. 4 ± 0.1	70 ± 2	3 ± 0.8	4 ± 1	12 ± 13
	SLPFs	Pasta (*n* = 32)	356 ± 7 *	0.5 ± 0.2 **	0.9 ± 0.3 **	0.6 ± 0.2 *	85 ± 3 **	0.6 ± 1 **	2 ± 3 **	59 ± 65 **
Rice	Regular food	Rice (*n* = 7)	351 ± 9	8 ± 0.8	12 ± 28	0.2 ± 0.2	77 ± 4	0.3 ± 0.5	2 ± 2	3 ± 2
	SLPFs	Rice (*n* = 7)	356 ± 9	0.4 ± 0.1 *	1 ± 0.2	0.5 ± 0.3 *	85 ± 4 *	0.1 ± 0.2	2 ± 2	22 ± 16
Cookies	Regular food	Cookies (*n* = 15)	475 ± 17	6 ± 0.7	19 ± 4	10 ± 3	68 ± 5	25 ± 8	3 ± 1	330 ± 127
	SLPFs	Cookies (*n* = 25)	497 ± 38 *	0.8 ± 0.5 **	22 ± 7	9 ± 5	73 ± 7 *	21 ± 9	1 ± 1 * (*n* = 24)	71 ± 67 **
Cakes/pancakes/pudding	Regular food	Cakes/pancakes/desserts (*n* = 15)	475 ± 17	6 ± 0.7	19 ± 4	10 ± 3	68 ± 4	25 ± 8	3 ± 1	330 ± 127
	SLPFs	Cakes/pancakes/desserts (*n* = 25)	497 ± 38 *	0.8 ± 0.5 **	22 ± 7	9 ± 5	73 ± 7 *	21 ± 9	1 ± 1 * (*n* = 24)	71 ± 67 **
Chocolate	Regular food	Chocolate (*n* = 11)	528 ± 26	7 ± 1	29 ± 4	16 ± 4	59 ± 4	55 ± 5	5 ± 10	83 ± 43
	SLPFs	Chocolate (*n* = 11)	485 ± 164	1 ± 1 **	29 ± 16	17 ± 11	61 ± 15	41 ± 17 *	2 ± 3 (*n* = 9)	87 ± 107
Ready meals	Regular food	Ready meals (*n* = 23)	195 ± 87	8 ± 3	7 ± 5	2 ± 2	26 ± 23	2 ± 2	2 ± 2	381 ± 216
	SLPFs	Ready meals (*n* = 17)	210 ± 147	1 ± 1 **	6 ± 6	3 ± 5	38 ± 32	2 ± 2	2 ± 2 (*n* = 10)	532 ± 342
Meat	Regular food	Meat (*n* = 23)	185 ± 79	20 ± 4	11 ± 9	4 ± 4	1 ± 2	0.7 ± 2	0.0 ± 0.0	714 ± 884
	SLPFs	Meat replacers (*n* = 11)	260 ± 140	4 ± 2 **	17 ± 16	9 ± 12	20 ± 23 **	2 ± 2 **	13 ± 11 ** (*n* = 7)	950 ± 326 *
Fish/seafood	Regular food	Fish/seafood (*n* = 18)	115 ± 46	17 ± 5	4 ± 4	0.9 ± 0.7	2 ± 5	0.2 ± 0.3	0.0 ± 0.0	226 ± 260
	SLPFs	Fish/seafood replacer (*n* = 8)	137 ± 80	1 ± 0.8 **	5 ± 6	1 ± 1	22 ± 12 **	2 ± 1 **	6 ± 13 *	450 ± 478 †
Egg	Regular food	Egg (*n* = 6)	133 ± 41	12 ± 0.7	9 ± 4	2 ± 1	0.7 ± 0.4	0.4 ± 0.4	0.0 ± 0.0	143 ± 12
	SLPFs	Egg replacer (*n* = 6)	326 ± 69 *	0.9 ± 2 *	0.4 ± 0.4 *	0.3 ± 0.3 *	71 ± 35 †	0.5 ± 0.8	18 ± 37 *	211 ± 224

^a^ The results are mean ± standard deviation. Data analysed using the Mann–Whitney test to compare regular foods and special low-protein foods (SLPFs): † *p* = 0.05–0.07; * *p* < 0.05; ** *p* < 0.001. ^U^: breakfast cereals included flavoured and plain cereals. CHO, carbohydrates; SFAs, saturated fatty acids.

**Table 2 nutrients-15-03475-t002:** Characteristics of study participants ^a^.

Characteristics	Participants with Disorders of Amino Acid Metabolism (*n* = 59) ^a^
Age, years (IQR)	8.9 (5.4–12.7)
Females, *n* (%)	27 (46)
Disorder of amino acid metabolism	
PKU, *n* (%)	30 (51)
PKU–BH_4_, *n* (%)	9 (15)
HCU, *n* (%)	4 (7)
HTI, *n* (%)	3 (5)
MSUD, *n* (%)	2 (3)
OAA, *n* (%)	4 (7)
MMA, *n* (%)	1 (2)
GA1, *n* (%)	1 (2)
3-HMG, *n* (%)	2 (3)
UCD, *n* (%)	4 (7)
ASS, *n* (%)	2 (3)
OTC, *n* (%)	1 (2)
HHH, *n* (%)	1 (2)
UCD, OAA liver transplanted, *n* (%)	3 (5)
ASL, *n* (%)	2 (3)
PA, *n* (%)	1 (2)
Anthropometric Parameters	
Weight Z-score (IQR)	0.9 (−0.7–0.7)
Height Z-score (IQR)	−0.7 (−1.8–0.2)
BMI Z-score (IQR)	0.6 (−0.4–1.3)
BMI Z-score −1 to 1 (between percentile 25 and 75), *n* (%)	35 (59)
BMI Z-score (>2), *n* (%)	7 (12)
BMI Z-score (1.5–1.95), *n* (%)	2 (3)
BMI Z-score (1.0–1.49), *n* (%)	9 (15)
BMI Z-score (−1.0–−1.49), *n* (%)	1 (2)
BMI Z-score (−1.5–−1.95), *n* (%)	3 (5)
BMI Z-score (<−2), *n* (%)	2 (3)
Plasma Biochemical Analysis (mg/dL)	
Glucose, mg/dL (IQR)	85 (81–92)
Normal glucose levels (<100 mg/dL), *n* (%)	55 (93)
Prediabetes risk glucose levels (100–125 mg/dL), *n* (%)	3 (5)
Type 2 diabetes risk glucose levels (>125 mg/dL), *n* (%)	1 (2)
Triglycerides, mg/dL (IQR)	67 (51–89)
Acceptable TAGs values (<75 mg/dL) (under 9 y), *n* (%)	20 (34)
Borderline TAGs values (75–99 mg/dL) (under 9 y), *n* (%)	5 (8.5)
High TAGs values (>100 mg/dL) (under 9 y), *n* (%)	5 (8.5)
Acceptable TAGs values (<90 mg/dl)(10–19 y), *n* (%)	23 (39)
Borderline TAGs values (90–129 mg/dL) (10–19 y), *n* (%)	4 (7)
High TAGs values (>130 mg/dL) (10–19 y), *n* (%)	2 (3)
Total Cholesterol, mg/dL (IQR)	136 (121–166)
Acceptable TC values (<170 mg/dL), *n* (%)	46 (78)
Borderline TC values (170–199 mg/dL), *n* (%)	9 (15)
High TC values (≥200 mg/dL), *n* (%)	4 (7)
HDL-C, mg/dL (IQR)	52 (47–60)
Acceptable HDL-C values (>45 mg/dL), *n* (%)	46 (77.9)
Borderline HDL-C values (40–45 mg/dL), *n* (%)	7 (11.9)
Low HDL-C values values (<40 mg/dl), *n* (%)	6 (10.2)
LDL-C, mg/dL (IQR)	76 (61–91)
Acceptable LDL-C values (<110 mg/dL), *n* (%)	50 (85)
Borderline LDL-C values (110–129 mg/dL), *n* (%)	7 (12)
High LDL-C values (≥130 mg/dL), *n* (%)	2 (3)

^a^ All values are in the median (IQR) or in *n* (%). 3-HMG, 3-hydroxy-3-methylglutaric aciduria; ASS, argininosuccinate synthetase deficiency; HCU, classical homocystinuria; ASL, deficiency of argininosuccinate lyase; GA1, glutaric aciduria type 1; HTI, hereditary tyrosinaemia type I; HDL-C, high-density lipoprotein cholesterol; HHH, hyperornithinaemia, hyperammonaemia and homocitrullinuria; LDL-C, low-density lipoprotein cholesterol; MSUD, maple syrup urine disease; MMA, methylmalonic acidaemia; OAA, organic acidaemias; OTC, ornithine transcarbamoylase deficiency; PKU, phenylketonuria; PKU-BH4, phenylketonuria with tetrahydrobiopterin treatment; PA, propionic acidaemia; TC, total cholesterol; TAGs, triglycerides; UCD, urea cycle disorders. Values for plasma lipid levels are from the NCEP Expert Panel on Cholesterol Levels in Children [28,29]. Values for plasma glucose and TAGs levels are from the Screening for Prediabetes and Type 2 Diabetes in Children and Adolescents, US Preventive Services Task Force Recommendation Statement [30].

**Table 3 nutrients-15-03475-t003:** Energy and macronutrient intake from the dietary data and stratified by special low-protein foods and supplement intakes of the 59 participants ^a^.

Variables	Dietary Intake (Including SLPFs and PFAAs) (*n* = 59)	SLPFs Intake (*n* = 59)	PFAAs Intake (*n* = 59)
Energy, kcal/day	1944 (1581–2432)	579 (341–813)	474 (278–800)
Energy, %	100	29 (20–36)	26 (14–35)
Total protein, g/day	56.1 (38.5–77.9)	1.9 (0.9–3.0)	40.3 (19.0–60.0)
Total protein, %	10.8 (8.2–14.2)	1.3 (0.8–2.0)	8.0 (3.8–11.3)
Natural protein, g/day	15.2 (9.9–21.1)	--	--
Natural protein, %	2.8 (1.9–4.5)	--	--
Carbohydrates, g/day	267.5 (202.7–343.9)	97.9 (67.6–154.9)	44.6 (21.1–81.9)
Carbohydrates, %	53.3 (50.7–57.3)	21.1 (15.1–27.8)	10.0 (3.8–13.6)
Sugars, g/day	65.8 (50.3–89.5)	13.0 (4.2–22.8)	6.5 (0.0–11.7)
Sugars, %	13.6 (10.2–16.5)	2.4 (1.2–4.0)	1.2 (0.0–2.7)
Fibre, g/day	19.0 (14.9–27.8)	2.5 (0.7–6.9)	0.5 (0.0–8.7)
Total fats, g/day	76.5 (63.7–88.7)	16.9 (7.0–22.5)	11.6 (2.8–20.7)
Total fats, %	34.0 (30.5–38.4)	7.6 (3.3–10.3)	5.5 (1.1–9.9)
SFAs, %	8.8 (7.3–11.7)	3.5 (1.8–4.7)	1.0 (0.0–2.0)
MUFAs, %	15.7 (12.6–17.5)	1.1 (0.1–2.2)	1.7 (0.0–4.4)
PUFAs, %	4.1 (3.1–5.2)	0.3 (0.1–0.5)	1.6 (0.0–2.6)
Cholesterol, mg/day	42.5 (6.8–85.4)	0.0 (0.0–0.2)	0.0 (0.0–0.0)

^a^ All values are in median (IQR). MUFAs, monounsaturated fatty acids; PFAAs, precursor-free L-amino acid supplement; PUFAs, polyunsaturated fatty acids; SFAs, saturated fatty acids; SLPFs, special low-protein foods.

**Table 4 nutrients-15-03475-t004:** Spearman’s correlation of % SLPF consumption.

	Energy from SLPFs, %	Carbohydrates from SLPFs, %	Sugars from SLPFs, %	Total Fats from SLPFs, %	SFAs from SLPFs, %
Total protein, %	0.09	0.14	0.03	−0.15	−0.16
Natural protein, %	−0.31 *	−0.49 **	−0.08	−0.04	−0.04
Carbohydrates, %	0.18	0.35 *	−0.05	−0.10	−0.13
Sugars, %	0.16	0.00	0.28 *	0.13	0.13
Total fats, %	−0.13	−0.28 *	0.07	0.26 †	0.27 *
SFAs, %	0.13	−0.08	0.25 †	0.39 **	0.52 **
MUFAs, %	−0.27 *	−0.45 **	−0.07	0.01	0.07
PUFAs, %	−0.37 **	−0.31 *	−0.21	−0.07	−0.07
Energy from SLPFs, %	--	0.88 **	0.72 **	0.72 **	0.66 **
Total protein from SLPFs, %	0.07	0.09	0.20	0.05	−0.01
Carbohydrates from SLPFs, %	0.88 **	--	0.56 **	0.54 **	0.43 **
Sugars from SLPFs, %	0.72 **	0.56 **	--	0.82 **	0.79 **
Total fats from SLPFs, %	0.72 **	0.54 **	0.92 **	--	0.91 **
SFAs from SLPFs, %	0.66 **	0.43 **	0.79 **	0.91 **	--
MUFAs from SLPFs, %	0.46 **	0.27 *	0.64 **	0.78 **	0.70 **
PUFAs from SLPFs, %	0.42 **	0.29 *	0.61 **	0.74 **	0.65 **
Energy from PFAAs, %	−0.26 †	−0.07	−0.21	−0.36 *	−0.30 *
Total protein from PFAAs, %	0.45 **	0.46 **	0.28 *	0.15	0.13
Carbohydrates from PFAAs, %	−0.29 *	−0.07	−0.29 *	−0.39 **	−0.33 *
Sugars from PFAAs, %	0.05	0.06	−0.08	−0.20	−0.18
Total fats from PFAAs, %	−0.38 **	−0.19	−0.30 *	−0.42 **	−0.34 *
SFAs from PFAAs, %	−0.44 **	−0.27 *	−0.37 **	−0.44 **	−0.35 *
MUFAs from PFAAs, %	−0.42 **	−0.28 *	−0.31 *	−0.45 **	−0.35 *
PUFAs from PFAAs, %	−0.31 *	−0.12	−0.26 *	−0.32 *	−0.25 †

* *p* < 0.05; ** *p* < 0.001; † *p* = 0.05–0.07.

**Table 5 nutrients-15-03475-t005:** Characteristics of study participants, dietary intake, and plasma biochemical parameters by tertiles of percentage of dairy SLPF consumption ^1^.

	Tertiles of Dairy SLPF Consumption			
	1 <1.0% (*n* = 19)	2 1.0–5.2% (*n* = 20)	3 >5.2% (*n* = 20)	*P*
Age, years	8.8 (7.4–14.6)	8.9 (4.2–14.5)	9.2 (5.7–11.4)	0.65
Females, *n* (%)	6 (31.6)	10 (37.0)	11 (40.7)	0.31
Anthropometric parameters				
Weight Z-score	0.31 (−0.41–0.55)	−0.39 (−0.71–0.99)	0.06 (−1.43–0.67)	0.79
Height Z-score	−1.03 (−2.02–0.16)	−0.61 (−1.81–0.72)	−0.69 (−1.50–0.16)	0.88
BMI Z-score	0.62 (−0.03–1.41)	0.84 (−0.29–1.31)	0.44 (−0.84–1.09)	0.36
Dietary intake				
Energy from SLPFs, %	23.9 (14.6–28.7) ^b^	30.2 (16.0–35.2) ^ab^	33.9 (25.7–43.5) ^a^	0.009
Energy from dairy SLPFs, %	0 (0–0) ^c^	2.8 (1.4–4.2) ^b^	10.7 (8.5–12.9) ^a^	<0.001
Energy from PFAAs, %	33.9 (20.0–36.7) ^a^	25.9 (14.4–29.4) ^ab^	20.4 (5.1–32.4) ^b^	0.041
Total protein, %	10.7 (9.3–16.0)	11.9 (8.3–14.0)	9.8 (7.7–13.8)	0.55
Natural protein, %	3.0 (1.9–5.1)	2.6 (1.9; 4.5)	2.8 (2.3–5.4)	0.59
Carbohydrates, %	56.1 (52.4–59.4)	53.3 (50.1–57.1)	52.8 (50.3–54.0)	0.08
Sugars, %	13.2 (8.5–16.5) ^b^	12.0 (9.4–13.9) ^b^	15.8 (14.0–17.7) ^a^	0.007
Total fats, %	31.9 (28.2–33.9) ^b^	34.9 (30.5–40.8) ^a^	36.8 (32.0–40.6) ^a^	0.012
SFAs, %	7.7 (6.1–8.6) ^c^	8.8 (7.1–11.4) ^b^	11.3 (9.5–12.7) ^a^	<0.001
MUFAs, %	14.3 (12.2–17.2)	16.0 (14.2–17.3)	16.9 (13.3–19.4)	0.20
PUFAs, %	4.6 (3.4–5.9)	4.1 (3.0–4.8)	3.9 (2.8–4.9)	0.24
Plasma biochemical parameters (mg/dL)				
Glucose, mg/dL	85.0 (83.0–94.0)	84.0 (77.0–90.5)	86.0 (80.3–93.5)	0.31
Triglycerides, mg/dL	67.0 (55.0–89.0)	60.5 (45.3–83.5)	71.5 (55.5–89.8)	0.39
Total cholesterol, mg/dL	131.0 (120.0–166.0) ^b^	125.0 (117.0–153.3) ^b^	164.5 (146.3–173.3) ^a^	0.006
HDL-C, mg/dL	52.0 (49.0–60.0) ^ab^	48.5 (44.3–52.8) ^b^	55.5 (49.3–67.3) ^a^	0.026
LDL-C mg/dL	67.0 (58.0–98.0) ^b^	69.5 (56.5–86.8) ^b^	87.0 (76.5–115.5) ^a^	0.011

^1^ All values are median (IQR) or *n* (%). Medians in a row with superscripts without a common letter differ, *p* < 0.05. Data analysed by Kruskal–Wallis and Mann–Whitney post hoc test or Chi-square test.

## Data Availability

The data sets generated and analysed during the current study are available from the corresponding author upon reasonable request.

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
