# Peer review of "Effect of Special Low-Protein Foods Consumption in the Dietary Pattern and Biochemical Profile of Patients with Inborn Errors of Protein Metabolism: Application of a Database of Special Low-Protein Foods"

_nutrients, 2023, doi:10.3390/nu15153475_

Round 1

Reviewer 1 Report

The title of this article is “Effect of special low-protein foods consumption in the dietary pattern and biochemical profile of patients with inborn errors of protein metabolism: Application of a database of special low-protein foods”. This is an interesting topic, and it is an area that needs our attention. However, there are still some areas of the article that need to be revised:

1.      Authors need to double check the writing of their articles, for example, the P-values in the abstract section of the article, where some are italicized and some are not, and authors need to align this.

2.      The article begins by identifying dietary modification as the primary treatment for IEM and suggests ways to supplement SLPFs. However, the article lacks information on the dangers of high-protein foods for IEM patients, for which the authors need to make appropriate additions to make the article more complete.

3.      The "Materials and Methods" section can be inserted into the article before the results to make it easier for readers to understand the article and increase the sense of hierarchy.

4.      In the discussion of the article, it is often pointed out that the results obtained from the experiment are similar to the results obtained from similar experiments in the past. In this regard, the authors should have discussed the experimental results in more depth, and the innovations of the experiments need to be pointed out more clearly and analyzed.

5.      Article 532, 533 lines. The authors mention the benefits of dietary fiber supplementation. For this part, the authors could have discussed in more depth from several perspectives, such as starting from the perspective of intestinal flora and further analyzing the benefits of mountain dietary fiber for the IEM population in the context of protein metabolism and the regulatory effects of dietary fiber on intestinal flora.

6.      The article provides a new idea for the treatment of IEM. However, the article lacks some revelation, and the authors need to give more of their own perspective and an outlook for the future in the context of practice.

7.      Authors are requested to carefully check the format of the references used in the article to ensure that the references are in the required format.

Please revise the English expressions in the manuscript by removing unnecessary "the" from the sentences, making sure the sentences look more concise, and replacing words that appear too often in the text.

Author Response

RESPONSES TO REVIEWER #1

The title of this article is “Effect of special low-protein foods consumption in the dietary pattern and biochemical profile of patients with inborn errors of protein metabolism: Application of a database of special low-protein foods”. This is an interesting topic, and it is an area that needs our attention. However, there are still some areas of the article that need to be revised:

Response: We thank the reviewer for his/her feedback. We appreciate his/her interest in the topic, and we agree that it is an area that requires attention. We will carefully review the identified areas that need revision to improve the article's quality.

Comment 1: Authors need to double check the writing of their articles, for example, the P-values in the abstract section of the article, where some are italicized and some are not, and authors need to align this.

Response 1: The referee is right in pointing out that P-values are not standardized. Consequently, we have now standardized and aligned them throughout the manuscript. Moreover, we have reviewed the writing, and the manuscript has undergone professional revision by an English editor.

Comment 2: The article begins by identifying dietary modification as the primary treatment for IEM and suggests ways to supplement SLPFs. However, the article lacks information on the dangers of high-protein foods for IEM patients, for which the authors need to make appropriate additions to make the article more complete.

Response 2: We agree with the referee’s comment. We have included the following statement in page 2, lines 51-62.

“The dietary treatment involves restricting natural protein intake contained mainly in foods such as meat, fish, eggs, dairy products, legumes, nuts and also in cereals, pasta and bread [4]. The amount of natural protein is determined individually based on the severity of the condition, age, rate of growth, and metabolic status. Thus, it is essential to avoid and minimize the consumption of protein-containing foods in the diet of these children due to the detrimental effects associated with their intake. The consumption of such foods can lead to the accumulation of harmful substrates, thereby exacerbating the condition. For example, in the case of PKU, elevated levels of phenylalanine can occur, while patients with UCD may experience the accumulation of ammonium. These metabolic imbalances can lead to progressive neurological deterioration and manifest various symptoms, such as lethargy, seizures, headaches, vomiting, and growth failure [2,5,6] “.

Comment 3: The "Materials and Methods" section can be inserted into the article before the results to make it easier for readers to understand the article and increase the sense of hierarchy.

Response 3: The “2. Materials and Methods” section (page 3) was included before the “3. Results” section (page 4).

Comment 4: In the discussion of the article, it is often pointed out that the results obtained from the experiment are similar to the results obtained from similar experiments in the past. In this regard, the authors should have discussed the experimental results in more depth, and the innovations of the experiments need to be pointed out more clearly and analyzed.

Response 4: As suggested by the reviewer, we have discussed the experimental results in more depth. Moreover, we have pointed out and analysed more clearly the innovations of our results. These changes have led to having to reduce and reorganize some parts of the discussion to focus on the main results and their discussion.

Comment 5: Article 532, 533 lines. The authors mention the benefits of dietary fiber supplementation. For this part, the authors could have discussed in more depth from several perspectives, such as starting from the perspective of intestinal flora and further analyzing the benefits of mountain dietary fiber for the IEM population in the context of protein metabolism and the regulatory effects of dietary fiber on intestinal flora.

Response 5: We agree with the referee’s comment that this section required a more comprehensive discussion.  Consequently, we have included the following statement in page 16, lines 488-496.

Despite the adequate fibre intake and its beneficial effects, Verduci et al., observed that in individuals with PKU, the low-protein diet led to an increase in carbohydrate intake, as well as higher glycemic index and glycemic load of their diet. This diet could result in a different quality of substrates for microbial fermentation, leading to reduced butyrate production and lower microbial diversity [44]. Bassanini et al., also found that the quality and quantity of carbohydrate ingested had an impact on the microbiome of PKU individuals, as they observed a depletion of the microorganism F. prausnitzii, which is considered a biomarker of health status and a butyrate producer [45].

Comment 6: The article provides a new idea for the treatment of IEM. However, the article lacks some revelation, and the authors need to give more of their own perspective and an outlook for the future in the context of practice.

Response 6: As suggested by the reviewer, we have incorporated our own perspective and a future outlook into the context of practice. These are predominantly presented in the following statements:

These results are consistent with our study, as we observed a higher contribution of dairy replacers to the diet of our patients (5 ± 6%) in comparison to SLPFs meat, fish, and egg re-placers (1 ± 2%). The variability in the nutritional composition of SLPFs highlights the need for up-to-date databases, which would greatly assist in optimizing individualized recommendations for children with IEM and protein restriction.” (page 15, lines 440-445).

“…. In light of these findings, it would have been highly insightful to explore whether individuals who consume higher quantities of SLPFs or dairy SLPFs exhibit distinct microbiota compared to those with lower consumption. Further investigations are necessary to delve deeper into this aspect.” (page 16, lines 507-510).

 “Having access to accurate and comprehensive nutritional information on SLPFs is crucial for healthcare professionals because they could provide effective dietary advice to patients and optimize dietary recommendations for patients, considering the potential impact of SLPFs on their overall health and well-being.” (page 16, lines 523-526).

Our study highlights the importance of regularly evaluating the protein-controlled diet in IEM in clinical practice to ensure adequate intake of SLPFs and prevent imbalances that could impact the lipid and glucose profile of these patients. As a result, they could prevent potential future complications, including cardiovascular risk factors.” (page 17, lines 546-550).

Comment 7: Authors are requested to carefully check the format of the references used in the article to ensure that the references are in the required format.

Response 7: As suggested by the reviewer, we have inserted the references in the required format.

Comments on the Quality of English Language: Please revise the English expressions in the manuscript by removing unnecessary "the" from the sentences, making sure the sentences look more concise, and replacing words that appear too often in the text.

Response: The manuscript has undergone editing by a certified English editor. Furthermore, we have replaced words that appeared excessively throughout the text.

Reviewer 2 Report

This study is very great for readers of Nutrients. I have major comments. Please, see below:

Introduction: What is hypothesis of study?

Methods: What is the design of study?

What is sample size calculus?

Results: Plasma biochemical analysis could be divided in high vs normal glicemia, high vs normal cholesterol HDL and LDL according to dairy SLPFs consumption.

Discussion: Is too long!!! and too many references were cited. Please, to reduce it. In addition, to focus on quality of the diet.

Conclusion: The last paragraph of conclusion may be removed.

Author Response

RESPONSES TO REVIEWER #2

This study is very great for readers of Nutrients. I have major comments. Please, see below:

Response: We thank the reviewer for his/her positive feedback. We are delighted to hear that you find our study valuable for readers of Nutrients.

Comment 1: Introduction: What is hypothesis of study?

Response 1: The hypothesis of the present study is: “A high consumption of special low-protein foods may potentially impact the diet quality in individuals with inborn errors of metabolism (IEMs) and could lead to alterations in their lipid and glucose profiles. Such changes in lipid and glucose profiles might act as triggering factors for future cardiovascular diseases in this specific population.”

We have included this hypothesis in the introduction of the manuscript (page 3, lines 105-109).

Comment 2: Methods: What is the design of study?

Response 2: We thank the reviewer for his/her comment. This is a cross-sectional observational study. We have included the design of the study in the abstract and in the “2.2 Subjects and study design” section (page 3, line 137).

Comment 3: What is sample size calculus?

Response 3: In this study we included all available patients diagnosed with IEMs managed at the Sant Joan de Deu Hospital during January 2021 and March 2022. It is to note that the different IEMs included in our work have low prevalence in populations and they are rare metabolic disorders. For instance, in the following lines we have collected the prevalence of the different IEMs included in our study:

  • Prevalence of PKU: In Spain, the overall prevalence of PKU is about 1/10,000 (Hillert et al., Am J Hum Genet. 2020; 107(2): 234–250).
  • Prevalence of MSUD: An estimated prevalence of MSUD is about 1/185,000 worldwide, however in certain communities there is an over-expression such as Galician community (North-West of Spain) that is 1/52,541 (Couce et al., European Journal of Paediatric Neurology. 2015; 19 (6): 652-659).
  • Prevalence of HCU: 1/200,000 births worldwide (Al-Sedeq & Nasrallah. Genes 2020, 11(3), 330).
  • Prevalence of organic acidaemias: The pooled prevalence of methylmalonic acidemia worldwide is 1.14/100,000 (Jin et al., J Matern Fetal Neonatal Med. 2022 Dec;35(25):8952-8967). Glutaric aciduria Type I about 1/75,000 in USA (Vaidyanathan et al., Indian J Clin Biochem. 2011; 26(4): 319–325).
  • Prevalence of urea cycle disorders: The overall average birth prevalence of UCDs is approximated to be 1/35,000 (Summar et al., Mol Genet Metab. 2013; 110(0): 179–180).
  • Prevalence of Tyrosinaemia type I: 1/100,000 (Äärelä et al., Orphanet J Rare Dis 2020; 15, 281).

Several publications have already stated that rare disease trials have smaller sample sizes than non-rare disease trials and studied the mean sample size considering the prevalence of IEMs: mean of 33.8 (95% CI, 22.1–51.7) for IEMs with 1-9/100,000 and mean of 35.6 (96% CI, 23.3–54.3) for IEMs with 1–5/10,000 (Hee, S.W., Willis, A., Tudur Smith, C. et al. Does the low prevalence affect the sample size of interventional clinical trials of rare diseases? An analysis of data from the aggregate analysis of clinicaltrials.gov. Orphanet J Rare Dis 12, 44 (2017)). In this sense, the cohort included in our work is the available population. However, in order to properly answer the reviewer’s comment, we have carried a post hoc power analysis (Faul et al. Behavior Research Methods. 2009; 41:1149–60; Higgins et al. Chapter 6. Choosing effect measures and computing estimates of effect. Cochrane Handbook for Systematic Reviews of Interventions. 2022): For total cholesterol assessed through dairy SLPF consumption, the estimated effect size of f = 0.45, alpha level of 5% for an ANOVA, one-way F-test and sample sizes n1 = 19, n2 = 20 and n3 = 20 revealed a statistical power of 86%. For LDL-C assessed through dairy SLPF consumption, the estimated effect size of f = 0.43, alpha level of 5% for an ANOVA, one-way F-test and sample sizes n1 = 19, n2 = 20 and n3 = 20 revealed a statistical power of 83%.

To account for all of that, we have included the following statements in the manuscript:

  • The eligibility of participants has been better explained in the manuscript in the 2.2 section: “Eligible patients for this study were children diagnosed with IEMs by newborn screening or genetically confirmed, treated by protein restricted diet, aged 10 months to 17 years, without language barrier or difficulty of understanding and managed at Con-genital Metabolic Disease Unit of Sant Joan de Déu Hospital (HSJD), as well as they attended clinical continuously appointments. In total, 59 children (27 girls and 32 boys) were recruited.” (page 3, lines 138-142).
  • We have added the following sentence at 2.4 Statistical Analysis (page 4): “In addition, post hoc power analysis was used to calculate the statistical power of total cholesterol and LDL-C parameters according to tertiles of dairy SLPFs consumption (Faul et al. Behavior Research Methods. 2009; 41:1149–60; Higgins et al. Chapter 6. Choosing effect measures and computing estimates of effect. Cochrane Handbook for Systematic Reviews of Interventions. 2022).
  • We have also added the following sentence at the discussion section (page 17): “The sample size is also limited, however, the statistical power for the main results is adequate since the estimated effect size for plasma total cholesterol levels assessed through dairy SLPF consumption revealed a statistical power of 86%. In addition, the estimated effect size for LDL-C assessed through dairy SLPF consumption revealed a statistical power of 83%.”

Comment 4: Results: Plasma biochemical analysis could be divided in high vs normal glicemia, high vs normal cholesterol HDL and LDL according to dairy SLPFs consumption.

Response 4: As suggested by the reviewer, we have now incorporated an additional Supplemental Table (Table S6) that presents the results of plasma biochemical analysis categorized into high, borderline and normal levels for glycemia, cholesterol, triglycerides, HDL-C and LDL-C according to tertiles of dairy SLPFs consumption. Furthermore, due to the significance of these findings, we have included a brief paragraph in the results section to highlight their relevance (page 14, lines 398-403).

Comment 5: Discussion: Is too long!!! and too many references were cited. Please, to reduce it. In addition, to focus on quality of the diet.

Response 5: As suggested by the reviewer, we have condensed the discussion section and removed some references. However, the other reviewer recommended a more comprehensive discussion of the experimental results and emphasized the innovations of our work. Consequently, while we have reduced the discussion compared to the initial version, we have enhanced the content concerning the quality of the diet to address the reviewer's input.

Comment 6: Conclusion: The last paragraph of conclusion may be removed.

Response 6:  We have revised and rephrased the final paragraph of the conclusion to provide a clearer explanation of how our results can be applied in clinical practice.

Round 2

Reviewer 2 Report

No more comments